# Role of Sex Hormones in Prevalent Kidney Diseases

**DOI:** 10.3390/ijms24098244

**Published:** 2023-05-04

**Authors:** Carolina Conte, Giulia Antonelli, Maria Elena Melica, Mirko Tarocchi, Paola Romagnani, Anna Julie Peired

**Affiliations:** 1Department of Experimental and Clinical Biomedical Sciences “Mario Serio”, University of Florence, 50134 Florence, Italy; 2Nephrology and Dialysis Unit, Meyer Children’s Hospital IRCCS, 50139 Florence, Italy

**Keywords:** chronic kidney disease, end-stage kidney disease, sex hormones, lupus nephritis, diabetic kidney disease, IgA nephropathy, autosomal dominant polycystic kidney disease, animal models

## Abstract

Chronic kidney disease (CKD) is a constantly growing global health burden, with more than 840 million people affected worldwide. CKD presents sex disparities in the pathophysiology of the disease, as well as in the epidemiology, clinical manifestations, and disease progression. Overall, while CKD is more frequent in females, males have a higher risk to progress to end-stage kidney disease. In recent years, numerous studies have highlighted the role of sex hormones in the health and diseases of several organs, including the kidney. In this review, we present a clinical overview of the sex-differences in CKD and a selection of prominent kidney diseases causing CKD: lupus nephritis, diabetic kidney disease, IgA nephropathy, and autosomal dominant polycystic kidney disease. We report clinical and experimental findings on the role of sex hormones in the development of the disease and its progression to end-stage kidney disease.

## 1. Sex Disparities in CKD

According to the latest epidemiologic studies, chronic kidney disease (CKD) affects more than 840 million persons worldwide [1]. Over the past 30 years, its death rate has increased by about 40%, and forecasting models predict that it will become the 5th most-common cause of potential years of life lost within the next 20 years [1,2]. These numbers can be partially explained by the rise in the main risk factors for CKD, obesity, and diabetes, in developed countries [1] as well as in low-to-middle-income countries [3]. CKD consistently shows a strong sexual dimorphism, with a higher age-standardized prevalence in females than in males despites geographical differences: 9.6% versus 8.6% in high-income countries, and 12.5% versus 10.6% in low-to-middle-income countries [4]. Nonetheless, the likelihood of progressing to end-stage kidney disease (ESKD) is higher in males [5,6]. The Chronic Renal Insufficiency Cohort (CRIC) study, which is a longitudinal, ongoing, multicenter study that involves almost 5500 adults with CKD in the United States (US), has revealed that women experience lower rates of kidney failure (defined as the need for dialysis or a kidney transplant) compared to men, with rates of 3.1 and 3.8 per 100 person-years, respectively [7,8]. After adjusting for sociodemographic, clinical, and laboratory characteristics through multivariate regression analysis, it was found that women had a decreased risk of ESKD, 50% estimated glomerular filtration rate (eGFR) decline, progression to CKD stage 5, and death compared to men [7,8].

The sex disparities revealed by CKD epidemiological data suggest a role for sex hormones, more specifically a protective role for estrogens. However, not all of the aforementioned studies have data on the menopausal state—and estrogen level—of the women included. When attention is made to separate premenopausal from postmenopausal women, the difference is clear, as in the Korean National Health and Nutrition Examination Survey (NHANES) from 2010, where the prevalence of CKD was 7.4% in men, 4.7% in premenopausal women, and 20.1% in postmenopausal women [9]. Analysis of the NHANES 1999–2014 data indicated an association between early menopause and CKD prevalence [10] (Figure 1). Indeed, women experiencing early natural menopause—before 45 years of age—were at a higher risk to develop CKD, while early surgical menopause was a hazard factor for both CKD and survival [10,11]. A long reproductive life span duration is also associated with a lower risk for CKD development, suggesting a cumulative protective effect of estrogen exposure over time [12,13]. However, women suffering from CKD often experience menstrual abnormalities and shorter reproductive life spans, making it difficult to diagnose menopause in premenopausal-age-group women with advanced CKD [14,15,16]. While postmenopausal hormone replacement therapy (HRT) slows down CKD progression [17], one should be wary of possible side effects of long-term treatment, such as coronary heart disease, venous thromboembolism, and stroke [18]. A meta-analysis and systematic review of the literature on transgender persons subjected to gender-affirming hormone therapy reveals that kidney function (as a change in serum creatinine) worsened in transgender men and improved in transgender women one year after the start of treatment [19]. Free testosterone levels are overall lower in male CKD patients and inversely correlated with the stage of the disease [20,21,22]. Reports have shown that testosterone levels increased in male patients after kidney transplant, that is, following an improvement in kidney function [23]. Low testosterone levels constitute a predictive factor for all-cause mortality in male dialysis patients [24]. Sex hormone levels in women are more difficult to assess due to their cyclic nature. Testosterone levels are not significantly different in women with or without CKD [25]. CKD women that underwent a successful kidney transplant had normalized serum concentrations of hormones linked to fertility disorders [26].

Animal models of CKD reflect the sex disparity observed in patients, with males progressing faster to CKD than females [6]. In 5/6 nephrectomy, male rats developed a more severe kidney phenotype, including albuminuria, anemia, and malnutrition, compared to females [27]. In chronic inhibition of nitric oxide, a model that exacerbates CKD progression, male rats displayed marked albuminuria, histological damage, interstitial inflammation, and tubulointerstitial fibrosis, while females had a milder phenotype [28]. In male rats with chronic allograft nephropathy following kidney transplant, testosterone treatment was detrimental, increasing inflammation and glomerulosclerosis, while estrogen treatment mitigated these alterations [29]. Similarly, estrogens protected from CKD progression in uninephrectomized rats [30], postmenopausal rats [31], 5/6 nephrectomized rats [32], kidney damage following acute kidney injury [33], adenine-induced CKD [34], and aging Dahl salt-sensitive rats [35].

Some kidney diseases that can cause CKD have been shown to have a sexual dimorphism or differences in prevalence, severity, or presentation between males and females (Table 1). This review will focus on the effect of sex hormones and their receptors on kidney physiology in selected prominent kidney diseases: lupus nephritis, diabetic kidney disease, IgA nephropathy, and autosomal dominant polycystic kidney disease.

## 2. Sex Hormones and Sex Hormone Receptors

Sex hormones are steroid hormones, including estrogens, progestogens, and androgens, that traditionally have been defined by their role in normal reproductive function, but that are also involved in other physiological and pathological processes. Sex hormones are produced by the gonads (ovaries or testes), by adrenal glands, or can be derived by conversion from other sex steroids in other tissues such as the liver or fat [59,60]. They may act in a paracrine manner or circulate in the blood, stabilized by the sex hormone-binding globulin (SHBG) or by albumin, or in a free form to act at the target tissue level in an endocrine fashion [60]. Sex hormones act via specific receptors in target tissues where they may exert their functions by slow transcription-dependent mechanisms through the cytoplasmic estrogen receptors (ER), androgen receptors (AR), and progesterone receptors (PR), as well as by fast transcription-independent mechanisms through membrane-associated receptors and signaling cascades [61,62,63]. For each sex hormone receptor, two forms exist: the estrogen receptors ERα and ERβ (encoded by the *ESR1* and *ESR2* genes, respectively) [64,65]; the androgen receptors AR-A and AR-B (two isoforms of the *AR* gene) [66], and the progesterone receptors PR-A and PR-B (two isoforms of the *PGR* gene) [67]. The selective expression of these isoforms in target tissues and the differential affinity for their ligands are at the basis of different cellular responses.

Blood circulating sex hormones can diffuse though the plasma membrane and, in the cytoplasm, bind to the specific receptor which undergoes dimerization (they can form both homodimers and heterodimers) and a consequent conformational change promoting the coupling to regulator proteins. These hormone-receptor complexes, in turn, translocate to the nucleus in which they bind to a hormone response element (HRE), a short sequence of DNA within the promoter of a gene, and therefore regulate transcription [62,63]. This signaling pathway takes several days to exhibit its response effects. Nevertheless, sex hormones can induce fast responses by transcription-independent mechanisms through the membrane-associated receptors: sex hormones can bind to membrane-associated receptors among which, in addition to the membranous-bound form of the previously mentioned receptor, are also the G protein-coupled estrogen receptor (GPER), which promote intracellular second messenger signaling, via MAPK/ERK/PI3K/cAMP signaling, that indirectly modulate gene expression and facilitates rapid changes [61,62,63].

Sex hormones and their receptors have important roles in normal kidney biology, and altered expression and function could explain the differences in kidney disease onset and progression.

## 3. Experimental Models

Acknowledging the sex-related disparities witnessed in experimental models of kidney disease is crucial when designing and interpreting research, as well as when creating novel therapies. Female mice, particularly in the C57BL/6J strain, have been shown to possess greater resistance to Adriamycin-induced injury [68,69]. The International Society of Nephrology (ISN) recently released a consensus guidance for preclinical animal studies in translational nephrology taking these distinctions into account. The ISN recommends that both sexes be studied to comprehend the pathophysiological differences linked to sex chromosomes, reproductive organs, and sex hormones in experimental models of kidney diseases [70].

A classical experimental model used to study the role of sex in preclinical studies is the gonadectomy, that is, the removal of the ovaries in females, or ovariectomy, and the removal of the testis in males, also called castration or orchiectomy. Gonadectomy can be performed in rats or mice, and can be combined with regular or cross-sex hormone therapy. Several animal models have been developed to mimic human menopause as well, from natural aging to ovariectomy and chronic exposure to ovotoxin [71]. The main limitation in all these models is the absence of specificity for a given hormonal pathway or hormone receptor.

Our current understanding of the physiological and pathophysiological roles of sex hormones is due, in large part, to the generation of various sophisticated genetic models of sex hormone receptor insufficiency, including ERα- and ERβ- deficient mice [72,73,74] and mice lacking androgen [75] or progesterone receptors [76]. Despite numerous animal models in which gonadal hormones or sex hormone receptors have been manipulated, to date, none have been accurately characterized for renal endpoints, although interest in the kidney research community has gradually increased in recent years. As many germline loss-of-function mutations in the sex hormone receptors in mouse models lack renal phenotypes, it is conceivable that abnormalities may only become evident under physiological stress, with age, or following kidney injury. The identification of numerous promoters conferring renal cell-specific gene regulation in vivo has greatly facilitated the interpretation of gene targeting studies [77,78]. In addition to the spatial delimitation driven by the promoter, temporal gene expression can be achieved using inducible systems, allowing the study of the effect of the absence of a hormone receptor during a specific stage of embryonic kidney development or in adult mice, for example [79,80,81]. Investigating renal cell type-specific sex hormone receptor knock-out in both female and male animals will be crucial to understanding sex differences attributed to sex hormone responsiveness in various kidney diseases. Furthermore, conducting similar genetic studies targeting known activators and repressors of sex hormones could lead to a better understanding of their role in cellular response.

The “four core genotypes” (FCG) mouse model is a valuable model for studying sex hormones, as it involves mice in which the sex chromosome complement is unrelated to the animal’s gonadal sex [82]. This model comprises XX and XY gonadal males or females, achieved through the deletion of the testis-determining gene Sry from the Y chromosome and the insertion of a Sry transgene onto an autosome [83,84]. The FCG model can distinguish between the influence of sex hormones and the effects of sex chromosomes. By combining gonadal hormones and sex hormone receptor deletions with existing experimental models of renal disease, it may be possible to develop valuable tools for studying sex differences. Additionally, removing the gonads of adult animals can eliminate sex differences in phenotypes caused by the distinct hormones produced by male and female gonads throughout their lifetimes.

To date, the insights that have been gained into human sex hormone deficiency are mostly due to the use of knock-out mouse models, even if the strengths and limitations of these mouse models should be considered to ensure an accurate interpretation of the phenotypes.

## 4. Lupus Nephritis

Lupus erythematosus (SLE) is an autoimmune disease that often causes severe kidney manifestations, called lupus nephritis (LN), which can progress in CKD and ESKD [36]. LN is a type of glomerulonephritis that can be classified based on mesangial involvement, presence of segmental or global sclerotic lesions, membrane thickening, or severity of tubulointerstitial lesions [85,86]. Evidence from clinical and experimental data highlights the sex disparities and the role of sex hormones in SLE and LN. Recent epidemiological studies show that in the US, the SLE incidence rate per 100,000 person-years was 5.1 (95% CI 4.6 to 5.6), and was higher in women than in men (8.7 vs. 1.2) [87]. Incidence varies greatly across international regions, but overall women are consistently more affected by SLE than men [88]. Up to 60% of SLE patients will develop LN in the course of their life, with a tendentially higher prevalence in male (27–75%) than in female patients with SLE (16–52%) [36]. In chronic graft-versus-host disease (GVHD), an experimental model of SLE, female mice were more susceptible to the development of glomerulonephritis [89,90]. Numerous studies demonstrated how sex hormones influenced the pathogenesis and clinical features of SLE (recently reviewed in [91]), advocating for the consideration of sex differences in the management of the disease. In particular, HRT and oral contraceptives can increase the susceptibility to developing SLE or to SLE flares [91,92,93]. In transgender women undergoing sex reassignment, gender-affirming hormone therapy can associate with SLE [94]. Treatment with 17beta-ethinyloestradiol in GVHD or NZB/WF1 mice—another lupus-prone model—accelerated autoantibody production and progression of LN [89,95,96]. Conversely, SLE mouse models deficient in ERα developed a mild phenotype [97,98], reinforcing the idea that estrogen is detrimental to SLE. In addition to estrogen, follicle-stimulating hormone (FSH), and luteinizing hormone (LH) could also affect SLE, as women with SLE experience disease flare during pregnancy, causing adverse outcomes [99]. In a murine model of SLE, pregnant females experienced a worsened kidney function, enhanced kidney inflammation, and a reduction in survival rate [100]. Androgen metabolism was found to have an important role in SLE, with SLE patients exhibiting lower levels of testosterone [93,101]. However, testosterone patches failed to provide any therapeutic benefit in SLE patients [102,103,104,105]. Whether progesterone plays a role in SLE patients is still debated [106]. In the NZB X NZW mouse model of SLE, females treated with medroxyprogesterone acetate—a synthetic form of progesterone used in oral contraceptives—have a lower mortality rate and serum IgG levels [107,108].

Sex hormones are known immunomodulators and have a prominent role in autoimmune diseases (reviewed in [109,110]). In particular, ER are expressed on a wide range of immune cells and promote the production of proinflammatory cytokines [111]. Modulation of cytokine expression by ER signaling in SLE patients as well as in experimental models of SLE has been reviewed elsewhere [112]. Estrogens have been shown to affect the development and function of B cells, T cells, and plasmacytoid dendritic cells (DCs) (Figure 2). Estrogens play an important role in B cell maturation, and ERα can increase autoantibody production [113]. Treatment with estradiol or interferon (INF) in NZB X NZW mice increased the expression of the B cell activating factor (BAFF)—a member of the TNF-family of proteins—in macrophages, and subsequent autoantibody production, which was reduced in ERα-deficient splenic cells [114]. An anti-BAFF monoclonal antibody, Belimumab, has been authorized by the US Food and Drugs Administration to treat lupus nephritis [115]. Upon estradiol stimulation, T cells of SLE patients start expressing the CD40 ligand, a known player in LN [116]. Importantly, in patients and mouse models of SLE, interleukin-2 (IL-2) production is impaired, resulting in a decrease in regulatory T cells (Treg)—a subset of CD4+ T cells that maintain self-tolerance by suppressing autoreactive lymphocytes—as well as excessive differentiation of CD4+ naïve T cells into proinflammatory T helper 17 (Th17) cells or T follicular helper (Tfh) cells, causing inflammation and autoantibody production, respectively. Patients with IL-2 deficiency often suffer from LN [117]. In mice models of SLE, estradiol treatment suppresses IL-2 signaling [118], while treatment with dehydroepiandrosterone (DHEA), an intermediate compound in testosterone synthesis, significantly upregulates IL-2 production [119]. An ongoing phase II clinical trial is assessing the efficacy of low-dose IL-2 therapy in patients with SLE [120]. Plasmacytoid DCs produce high levels of type I interferon (IFN) in SLE patients, through toll-like receptor (TLR-)-7 and TLR-9 [112]. Estrogen treatment increased TLR-mediated production of IFNα by DCs of postmenopausal women and of SLE-prone mice and restored it in Erα-deficient mice [121,122]. Despite important side effects, chloroquine is still commonly used to treat SLE, inhibiting TLR signaling [123].

Testosterone, conversely, is immunosuppressive, as it inhibits B cell lymphopoiesis in the bone marrow [124]. Accordingly, male mice deficient in the androgen receptor have increased numbers of bone marrow B cell precursors [125]. Recently, testosterone has been shown to regulate B cells through the cytokine BAFF, an essential survival factor for splenic B cells [126].

Progesterone has been shown to inhibit immune cell activation in SLE, as medroxyprogesterone acetate administration in female mice lowered anti-DNA IgG and CD86 on dendritic cells, but increased the expression of CD40 on B cells [108]. Conversely, aged Nba2 female mice deficient in the progesterone receptor had more anti-DNA IgG and more glomerular IgG deposition, inflammation, and overall injury [127]. These mice also had fewer splenic Treg cells and more Tfh cells [127].

Sex hormone influence on the immune system a major role in SLE, creating significant differences between male and female patients, increasing the complexity of drug design.

## 5. Diabetic Kidney Disease

Diabetic kidney disease (DKD), which impacts 30% of type 1 and 40% of type 2 diabetes mellitus (DM) patients, is a prevalent microvascular complication of the disease. DM is the leading cause of CKD worldwide, and it develops in approximately 40% of DM patients [128,129,130]. Epidemiological studies showed that males progress from DKD to CKD to ESKD faster than premenopausal women [45,46]. Sex disparities have been observed in the prevalence and incidence of DKD (diagnosed by the presence of albuminuria and low eGFR), as well as its phenotypes and clinical manifestations [45,46,47,131]. In a prospective observational study of 191 patients with type 2 diabetes mellitus who were monitored for a median of 5.8 years, male gender was found to be the second risk factor for the development of incipient or overt DKD, after albuminuria [47]. Additionally, an association between sex and incident DKD was noted in a follow-up of almost 10 years of 1464 patients with diabetes and normal renal function at baseline, leading to the conclusion that women with diabetes have a higher risk of incident CKD than men [132].

Although the exact mechanisms underlying this sex-based disparity remain unclear, sex hormones are known to play a significant role in the pathophysiology of diabetes and its complications, especially in women with DM who seem to lose the protective effects of estrogens on the cardiovascular system even before menopause (Figure 3). The loss of estrogen is a significant factor in explaining these differences [133]. Under healthy conditions, estrogens act as vasodilators, increasing the expression of nitric oxide synthase in the endothelium and resulting in phosphorylation and nitric oxide production via the ERα receptor [134]. Moreover, in animal models, estrogens appear to reduce fibrosis and apoptosis in the kidney [135]. Wells et al. demonstrated in a rat model of diabetes that estradiol supplementation may be an effective regimen in attenuating the onset and progression of diabetic renal complications [136]. Evidence suggests that, in human studies, estrogen therapy attenuates the progression of DKD. Maric et al. showed that supplementation with 17β-estradiol or administration of selective estrogen receptor modulators reduces the incidence of diabetes and attenuates the progression of DKD [133]. Estrogen replacement therapy decreases hyperandrogenicity and improves glucose homeostasis and plasma lipids in postmenopausal women with noninsulin-dependent DM [137]. Similarly, Brussaard et al. claimed that short-term estrogen replacement therapy improves insulin resistance, lipids, and fibrinolysis in postmenopausal women with non-insulin-dependent DM [138]. Szekacs et al. report that postmenopausal hormone replacement improves proteinuria and impaired creatinine clearance in type 2 diabetes mellitus and hypertension [139].

Additionally, in a mouse model of type 2 DM, the db/db mouse model, characterized by a phenotype of severe obesity, hyperphagia, polydipsia, and polyuria caused by a spontaneous mutation in the leptin receptor, using a selective estrogen reporter modulator, raloxifene, results in reduced albuminuria levels and renal damage [140]. Hadjadj et al. tested the hypothesis that raloxifene protects against increasing urinary albumin excretion in post-menopausal women with type 2 DM in a randomized pilot clinical trial. They observed that raloxifene may limit the progression of albuminuria [141]. Another drug that provides protection in DM is Vitamin D. Experimental data demonstrated that supplementation with Vitamin D or its active derivatives improves endothelial cell injury, reduces proteinuria, attenuates renal fibrosis, and as a result, retards DKD progression [142]. It is interesting to note that estrogens and Vitamin D have a bidirectional relationship, with estrogens interfering with Vitamin D immunomodulatory activities and vitamin D downregulating the aromatase effect [143,144]. Combining Vitamin D supplementation and sex steroid therapy appears to protect endothelial integrity and counteract cardiovascular damage that contributes to CKD and DKD progression. Oral estrogen supplementation has various mechanisms for its renoprotective effects in DM women [145,146]. In rat models, it has been shown that aldose reductase may both exacerbate and alleviate the production of metabolites that lead to hyperglycemia-induced cellular impairment. The assessment of oxidative stress in diabetic and hypertensive patients may also be a predictive factor for the progression of kidney damage [147]. E2 therapy can interfere with various pathways of glycemic damage, including the accumulation of Advanced Glycation End Products (AGEs) and the expression of transforming growth factor-beta (TGF β), AT1 receptors, and endothelins with a decrease in the production of collagen and a reduction in apoptotic phenomena [30,148,149]. In fact, another pathway of glycemic damage is represented by an estrogen-associated increase in the function of the receptor for advanced glycation end-products (RAGE). For this reason, under diabetic conditions, AGEs are excessively generated through the aldose reductase (AR)-polyol pathway. AGEs reduce the efficiency of anti-oxidant systems, downregulating several protective molecules, such as AGER1 and SIRT1 [150,151]. Estrogens may also increase the activity of nitric oxide synthase at the glomerular level, improving vascular permeability and glomerular function [152]. The regulation of TGF-β pathways is likely influenced by estrogen levels, with its expression upregulated by chronic hyperglycemia leading to glomerulosclerosis [153,154]. TGF-β levels are usually increased in men and reduced in women [155]. Estrogens have been reported to reduce the activity of RAAS (renin–angiotensin–aldosterone system) and stimulate TGF-β, confirming its role in regulating TGF-β pathways [156,157,158]. Progesterone also appears to play an important role in kidney protection, with progesterone receptors mainly located in the epithelial cells of the distal tubule in both male and female subjects [159]. Estrogen administration, alone or in addition to progesterone replacement therapy [160], has demonstrated beneficial effects on ischemic tubular damage [161]. Progesterone administration in ovariectomized diabetic mice has been shown to improve the outcomes of diabetic kidney disease, reducing glomerulosclerosis and profibrotic/angiogenetic factors and downregulating podocyte markers such as nephrin and podocin [160,162].

Regarding testosterone, some studies show that the decrease in testosterone levels and concomitant increase in estradiol and progesterone levels with DM correlate with the development of albuminuria, a hallmark of DKD [163,164]. Interestingly, castration in diabetic rats, which reduced testosterone levels even further than that in intact diabetic rats (but had no additional effect on either estradiol or progesterone), was associated with more severe albuminuria than that in intact diabetic rats. In addition, Iada et al. showed that adjusting the 17β-estradiol-to-androgen ratio ameliorates DKD in a mouse model [165,166,167]. Similarly, in rat models, inhibition of estradiol synthesis attenuated renal injury in male streptozotocin-induced diabetic animals [168]. Sex hormones, particularly estrogens, play a key role in the pathophysiology of diabetic renal disease. Although the evidence mentioned above has provided valuable information on the direct and indirect effects of estrogens in the kidney, further research is needed to clarify the relationship between sex hormones and the incidence and progression of diabetic renal disease.

## 6. IgA Nephropathy

Immunoglobulin A nephropathy (IgAN) is a common form of immune complex glomerulonephritis progressing to ESKD [169] The pathophysiology of IgAN is not clear, but a hypothesis involves the development of antibodies against aberrantly glycosylated O-linked oligosaccharide(s) on the IgA1 hinge region [170]. The accumulation of pathogenetic polymeric IgA1 immune complexes (occasionally with IgG and IgM) in the glomerular mesangium, leads to ESRD in 30–40% of patients within 20–30 years of diagnosis [171]. Interestingly, Nakamura et al. found that females had significantly higher antibody activity against synthetic hinge peptides and glycopeptides, which could suggest a protective mechanism in females against aberrantly glycosylated molecules [49].

The incidence and clinical manifestations of IgAN show gender differences, with male patients at a higher risk of developing ESKD and having worse outcomes than females [50,172]. This gender-based disparity has been observed in other kidney diseases as well, such as idiopathic membranous nephropathy and ADPKD, as reported in a meta-analysis by Neugarten et al. [44].

Additionally, sex-specific gene polymorphisms have been found to be associated with IgAN: the NTN4 rs1362970 A/A and GNG2 rs3204008 G/G genotypes are associated with increased IgAN risk in males, and the PHLDB1 rs7389 G/T genotype is associated with a higher risk in females [51].

Studies investigating the impact of sex on the development of ESKD in IgAN have produced varying results. While some studies have found no differences between males and females [50,52], others have reported worse outcomes in females [173]. For instance, although blood pressure was higher in males, proteinuria did not differ between the sexes at diagnosis or during follow-up evaluation [52]. Nevertheless, other studies have demonstrated more rapid eGFR decline, worse clinicopathological characteristics, and quicker disease progression in men [44,50]. In an animal study, B6C3F1 mice with vomitoxin- (VT)-induced IgAN exhibited a male predisposition and more severe disease outcomes [174]. To explore the role of estrogen in IgAN, a study found that female B6C3F1 mice who underwent castration showed increased severity of VT-induced IgAN, but estrogen supplementation did not mitigate this effect and instead increased disease severity [175].

Interestingly, some of the key genes upregulated in IgAN were linked with the estrogen signaling pathway and a polymorphism of the ERα gene might be associated with the pathogenesis of IgAN [176,177]. Another study reported that the expression of glomerular ERα in IgAN kidney tissue decreased with the worsening of the disease, proposing ERα as an independent factor involved in the prognosis of patients with IgAN [178]. However, further studies are needed to elucidate the roles of estrogen and androgen and their receptors in the pathogenesis of IgAN.

## 7. Autosomal Dominant Polycystic Kidney Disease

ADPKD is the most common monogenetic hereditary renal disorder in adults and it is caused by the mutation of PKD1 or PKD 2 [179]. ADPKD often results in CKD and ESKD.

Male gender is a risk factor for the progression of ADPKD [179]. Affected men present a rapid decline of renal function and earlier onset of end-stage renal disease, compared to women [55]. In a rat model of inherited polycystic kidney disease (PKD), orchiectomy led to a reduction in renal size and cyst volume density, indicating alleviation of renal disease [180]. In contrast, testosterone substitution was shown to antagonize the protective effect of gonadal ablation. Moreover, in females, testosterone increased kidney size and cyst growth, identifying androgens as a progression factor [180]. Conversely, estrogens have been proposed as protective hormones in some experiments with rats [135,181], where the authors showed that in males the presence of intact androgen status is associated with stimulation of the renin-angiotensin system (RAS) and endothelin-1 (ET-1) systems. Conversely, in females, estrogen has a protective effect, inducing suppression of the intrarenal RAS and ET-1 systems, and upregulation of VEGF, thereby promoting the preservation of renal function and attenuation of the loss of structure [135]. In conclusion, androgens and estrogens seem to have a role in cyst growth and disease progression in ADPKD.

Recent evidence has highlighted the vital role of two chloride ion channels, namely the protein kinase A-regulated cystic fibrosis transmembrane conductance regulator (CFTR) and the Ca^2+^-activated Cl- channel transmembrane 16A (TMEM16A), in the pathology of ADPKD [182]. It has been discovered that the TMEM16A promoter region contains androgen-response elements that are relevant for the testosterone-dependent induction of TMEM16A [183]. Despite the variations in TMEM16A expression, CFTR may be expressed at lower levels in female ADPKD patients, which could contribute to reduced renal cyst growth in females. This decrease in CFTR expression may be due to estrogen-dependent regulation of CFTR [184,185].

In a recent study, Talbi et al. investigated whether enhanced expression or function of TMEM16A and/or hormonal regulation may induce a more severe phenotype in male ADPKD patients [182]. The authors showed that the more severe cystic phenotype in men is likely to be caused by enhanced cell proliferation, possibly due to enhanced basal and ATP-induced intracellular Ca^2+^ levels, leading to enhanced TMEM16A currents. This study also suggests a difference in Ca^2+^ homeostasis in the kidneys of male and female ADPKD patients [182]. Further studies of specific gender hormone modulation will be essential for the development of clinically applicable approaches to slowing down the progression of ADPKD.

## 8. Conclusions and Future Directions

The sex-related discrepancies observed in clinical and experimental settings of CKD and prominent kidney diseases causing CKD can be explained in part by the effect of sex hormones on the kidney. Recent studies in complex diseases reveal the close relationship between sex hormones and epigenetic modifications, i.e., changes in gene expression patterns that are not directly related to the alteration of the DNA sequence itself [186,187]. Studies on epigenetic alterations in a mouse model of DKD have shown a sexual dimorphism in the expression of key (de)methylation enzymes [188], suggesting a possible link to differences in sex hormone expression between males and females. In CD4+ T cells from SLE patients, estradiol treatment inhibited DNA methyltransferase 1, causing global DNA hypomethylation [189]. While the role of epigenetic modifications in kidney diseases has been extensively studied [190], its link to sex hormones is still anecdotal and would require further investigation.

The advancement of new technologies such as transcriptomics and pharmacogenomics has provided insights into the sex-specific molecular mechanisms underlying kidney physiology. The study by Ransick et al. combined single-cell analyses with genetic fate mapping to show sex, lineage, and regional diversity in the mouse kidney, presenting the distribution of selected genes associated with hormonal regulation and allowing for further exploration of sexually dimorphic gene activity [191]. Transcriptomics studies [192], and more recently multiomics analyses [193], have revealed sex differences in mouse renal proximal tubular cells. In CKD, impaired function and activation of proximal tubular cells contribute to the development of tubulointerstitial fibrosis and renal tubular atrophy, leading to progressive loss of renal function. Therefore, knowing the sex-biased expression of key molecules in physiological conditions provides the basis for further studies in CKD conditions.

As the use of artificial intelligence and machine-learning approaches to analyze large datasets generated by these technologies could enable the development of predictive models for the early detection and personalized treatment of CKD, it is essential to create sex- and gender-sensitive models, as proposed, for example, in the field of kidney transplantation [194]. The development of sex-specific drug dosing and administration protocols, informed by sex-biased expression of pharmacogenes across human tissues, as described in the study by Idda et al., could also improve treatment outcomes in CKD patients [195].

Moving forward, the application of these technologies could facilitate the identification of sex-specific biomarkers of CKD progression and novel therapeutic targets and the development of more effective treatments for both male and female CKD patients.

## Figures and Tables

**Figure 1 ijms-24-08244-f001:**
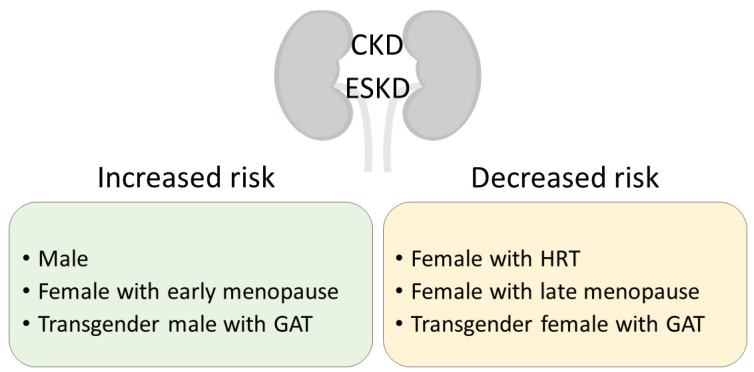
Clinical significance of sex in CKD development and progression to ESKD. CKD, chronic kidney disease; ESKD, end-stage kidney disease; HRT, hormone replacement therapy; GAT, gender-affirming therapy.

**Figure 2 ijms-24-08244-f002:**
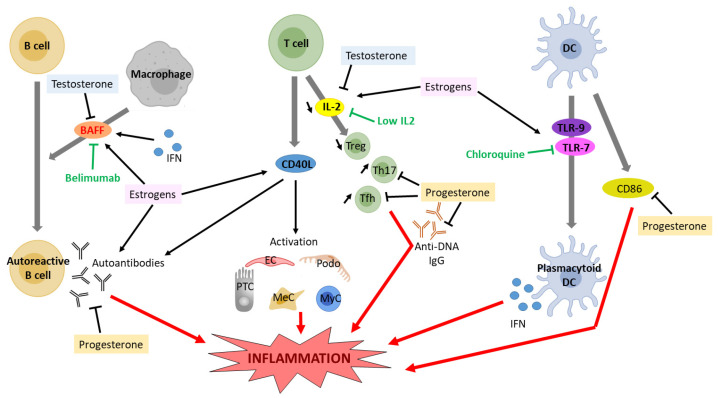
Effect of estrogen and testosterone signaling on cytokine expression and proinflammatory response in lupus nephritis. DC, dendritic cell; IFN, interferon; IL2, interleukin 2; BAFF, B cell activating factor; PTC, proximal tubular cell, MyC, myeloid cell; Podo, podocyte; MeC, mesangial cell; EC, endothelial cell; Treg, regulatory T cell; Th17, T helper 17 cell; Tfh, T follicular helper; TLR, toll-like receptor.

**Figure 3 ijms-24-08244-f003:**
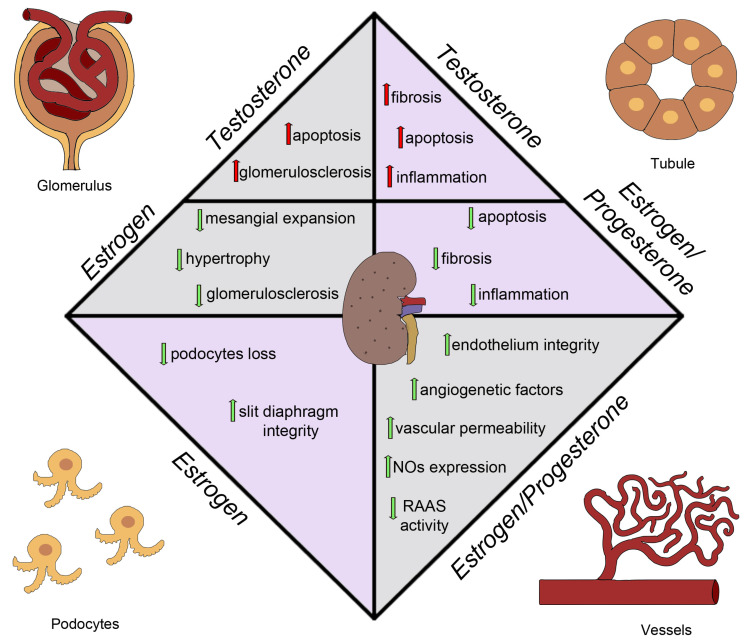
Sex hormones effects on diabetic kidney disease. NO, nitric oxide; RAAS, renin–angiotensin–aldosterone system.

**Table 1 ijms-24-08244-t001:** Sex-specific associations of kidney diseases.

Kidney Disease	Sex-Specific Associations
Lupus nephritis	➢ Higher prevalence in men with SLE [36] ➢ More aggressive histopathological features in men [37,38] ➢ Women more likely to achieve complete remission [39,40] ➢ No gender differences in long-term renal outcomes or mortality [38,41,42]
Minimal change nephropathy	➢ No gender differences in clinical phenotype or remission rates [43]
Diabetic kidney disease	➢ Faster progression from DKD to CKD to ESKD in men than in premenopausal women [8,44] ➢ Presence of albuminuria and low eGFR in men [45,46,47]
Anti-GBM disease	➢ Clinical features dependent on age and smoking status rather than gender [48] ➢ No gender differences in long-term renal outcomes [48]
IgA Nephropathy	➢ Higher antibody activity against aberrantly glycosylated IgA in women [49] ➢ Faster eGFR decline, adverse clinicopathological characteristics, and rapid disease progression in men [44,50] ➢ Sex-specific gene polymorphisms associated with increased risk in men [51] ➢ No gender differences in proteinuria, disease activity, or outcomes [50,52]
Focal segmental glomerulosclerosis	➢ Higher levels of proteinuria in men [52] ➢ Greater risk of relapse and less likely to attain remission in men [51,52,53] ➢ Increased risk of death in men [51,54]
Autosomal dominant polycystic kidney disease	➢ Rapid decline of renal function and earlier onset of ESKD in men [55]
ANCA associated vasculitis	➢ No gender differences in clinical outcomes [56,57,58]

SLE, systemic lupus erythematosus; DKD, diabetic kidney disease; CKD, chronic kidney disease; ESKD, end-stage kidney disease; eGFR, estimated glomerular filtration rate; GBM, glomerular basement membrane; ANCA, anti-neutrophil cytoplasmic antibody.

## Data Availability

Not applicable.

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
