# Peer review of "Role of Sex Hormones in Prevalent Kidney Diseases"

_ijms, 2023, doi:10.3390/ijms24098244_

Round 1

Reviewer 1 Report

The current manuscript describes the gender difference in clinical manifestations of chronic kidney disease (CDKs), specifically focusing on the effects of sex steroid hormones. The theme itself is not new, but very important for clinicians to understand the complicated pathophysiology of renal diseases causing CDK. This review covers representative kidney diseases such as lupus nephritis, diabetic kidney disease, IgA nephropathy, and autosomal dominant polycystic kidney disease.

The quality of the English language is good, and the Figures are concise and clear. Below are the issues the authors should consider revising.

  • It is helpful for readers if the reference numbers of articles that account for each reaction/effect are shown in Figure 2 and Figure 3, like in Table 1.
  • Lines 212 -213 (castrated mouse-model of SLE presenting delayed onset of disease [102] ); there seems to be no relevant information for this sentence in the reference [102].
  • Lines 220 (elicit a proinflammatory response trough cytokine expression); it is not easy to understand this sentence.
  • Lines 266-267; there seems to be no relevant information for this sentence in the reference [8]. The study clearly says, "no evidence of significant effect modification by diabetes on the association between sex and ESRD." The reference [44] is a study about the gender effect on the progression of nondiabetic renal disease."
  • Lines 272-274; the conclusion of this study [130] is "women with diabetes had a higher risk of incident CKD compared with men." It's better to state it clearly in the main text.
  • Line 295 (the db/db mouse model); please explain this model briefly in the main text.
  • Lines 308-313; the authors need to explain more about how estrogen is related to aldose reductase/oxidative stress in the main text.
  • Lines 322-323; the reference [152] seems irrelevant to hyperglycemia.
  • Lines 324-325; the reference [154] has no description of estrogen.
  • Lines 362-366; these sentences are linked to the previous description (lines 352-354). Please consider moving the sentences (lines 362-366) to the first paragraph (lines 351-357).
  • Lines 382-385; the description is inconsistent with the results shown in the references [170, 171].
  • Lines 395-399; the study in the reference [174] uses a rat as an animal model. Please state it clearly.
  • Lines 399 (On the other hand, estrogens have been proposed as protective hormones [175]); the reference [175] has no description of ADPKD.
  • Lines 412-414; this sentence is irrelevant to kidney diseases. The authors'd better delete or replace it with a sentence related to renal tissue/kidney diseases.

The quality of the English language is good. There are several minor typos in the text. 

Author Response

The current manuscript describes the gender difference in clinical manifestations of chronic kidney disease (CDKs), specifically focusing on the effects of sex steroid hormones. The theme itself is not new, but very important for clinicians to understand the complicated pathophysiology of renal diseases causing CDK. This review covers representative kidney diseases such as lupus nephritis, diabetic kidney disease, IgA nephropathy, and autosomal dominant polycystic kidney disease.

Thank you for taking the time to review our paper. We greatly appreciate your positive feedback and are thrilled to hear that you found our work to be good. Your comments have been valuable in helping us improve the manuscript and we are grateful for your insights and suggestions.

The quality of the English language is good, and the Figures are concise and clear. Below are the issues the authors should consider revising.

  • It is helpful for readers if the reference numbers of articles that account for each reaction/effect are shown in Figure 2 and Figure 3, like in Table 1.

We added the reference numbers to figure 2 and 3.

  • Lines 212 -213 (castrated mouse-model of SLE presenting delayed onset of disease [102] ); there seems to be no relevant information for this sentence in the reference [102].

We deleted this phrase.

  • Lines 220 (elicit a proinflammatory response trough cytokine expression); it is not easy to understand this sentence.

We modified this sentence.

  • Lines 266-267; there seems to be no relevant information for this sentence in the reference [8]. The study clearly says, "no evidence of significant effect modification by diabetes on the association between sex and ESRD." The reference [44] is a study about the gender effect on the progression of nondiabetic renal disease."

We replaced the references.

  • Lines 272-274; the conclusion of this study [130] is "women with diabetes had a higher risk of incident CKD compared with men." It's better to state it clearly in the main text.

Done.

  • Line 295 (the db/db mouse model); please explain this model briefly in the main text.

Done.

  • Lines 308-313; the authors need to explain more about how estrogen is related to aldose reductase/oxidative stress in the main text.

Done

  • Lines 322-323; the reference [152] seems irrelevant to hyperglycemia.

We replaced the reference.

  • Lines 324-325; the reference [154] has no description of estrogen.

We replaced the reference.

  • Lines 362-366; these sentences are linked to the previous description (lines 352-354). Please consider moving the sentences (lines 362-366) to the first paragraph (lines 351-357).

We moved the sentences.

  • Lines 382-385; the description is inconsistent with the results shown in the references [170, 171].

We modified this sentence.

  • Lines 395-399; the study in the reference [174] uses a rat as an animal model. Please state it clearly.

We modified this sentence.

  • Lines 399 (On the other hand, estrogens have been proposed as protective hormones [175]); the reference [175] has no description of ADPKD.

We replaced the reference.

  • Lines 412-414; this sentence is irrelevant to kidney diseases. The authors'd better delete or replace it with a sentence related to renal tissue/kidney diseases.

We deleted this sentence

The quality of the English language is good. There are several minor typos in the text. 

We did our best to find the typos and fix them

Reviewer 2 Report

This is a review on the relationship between gender (sex hormones) and kidney disease, and etiology and evidence are summarized for both animal examination and human. It is a solid and useful paper. 

I don’t think there is anything to fix.

Author Response

Thank you for taking the time to review our paper. We greatly appreciate your positive feedback and are thrilled to hear that you found our work to be good.

Reviewer 3 Report

In this paper, the authors sought to present a clinical overview of the gender-differences in CKD and a selection of prominent kidney diseases causing CKD: lupus nephritis, diabetic kidney disease, IgA nephropathy and autosomal dominant polycystic kidney disease. The authors report clinical and experimental findings on the role of sex hormones in the development of disease and  progression to end stage kidney disease. Overall, this is a timely review on a topic of great interest. I congratulate the authors on this interesting paper.

Author Response

(The authors gave the same response as above.)

Reviewer 4 Report

The authors reviewed pathophysiological roles of sex hormones in different types of kidney diseases. Since the manuscript adequately summarizes past research findings, the article will help potential readers understand importance of sex hormones in the development of kidney diseases or protection of kidney functions. Furthermore, the authors clarify current biological and/or medical questions which should be answered in future studies in the manuscript.

Minor comment

Recent studies have demonstrated that epigenetic sex differences also key mechanisms underlying pathological sex differences. Thus, it would be interesting to discuss contributions of epigenetic sex differences as well as sex hormones to the development of kidney diseases.

Author Response

Thank you for taking the time to review our paper. We greatly appreciate your positive feedback and are thrilled to hear that you found our work to be good.

Minor comment

Recent studies have demonstrated that epigenetic sex differences also key mechanisms underlying pathological sex differences. Thus, it would be interesting to discuss contributions of epigenetic sex differences as well as sex hormones to the development of kidney diseases.

We agree with the reviewer on the importance of epigenetic sex differences in the development of kidney disease, and discussed it in the paragraph “Conclusions and future directions”.

Round 2

Reviewer 1 Report

The authors answered all my comments.

Very good review, congratulation!